# Harnessing HIV clinics to deliver integrated hypertension care for People living with HIV in Uganda: A formative mixed methods study

Fred C. Semitala[1,2,3]☯, Florence Ayebare📷[3]☯*, John Baptist Kiggundu[3], Christine Kiwala[3], Joel Senfuma[3], Gerald N. Mutungi[4], Isaac Ssinabulya[1,5,6], James Kayima[5], Martin Muddu[3], Donna Spiegelman[7], Jeremy I. Schwartz[6,7], Chris T. Longenecker[8], Anne R. Katahoire[9]

1 Department of Medicine, Makerere University, Kampala, Uganda, 2 Makerere University Joint AIDS Program, Kampala, Uganda, 3 Infectious Diseases Research Collaboration, Kampala, Uganda, 4 Directorate of Non-Communicable Diseases, Ministry of Health, Kampala, Uganda, 5 Uganda Heart Institute, Kampala, Uganda, 6 Uganda Initiative for integrated Management of Non-Communicable Diseases, Kampala, Uganda, 7 Section of General Internal Medicine, Yale University, New Haven, Connecticut, United States of America, 8 Division of Cardiology, Department of Global Health, University of Washington, Seattle, Washington, United States of America, 9 Child Health and Development Centre, Department of Medicine, Makerere University, Kampala, Uganda

☯ These authors contributed equally to this work.
* fayebare@gmail.com

**Data availability statement:** Data reported in this manuscript are publicly available at Harvard Dataverse at the following url: https://doi.org/10.7910/DVN/Z1MLGQ.

**Funding:** The National Heart, Lung and Blood Institute (NHLBI) of the National Institutes

## Abstract

Access to antiretroviral therapy has led to better treatment outcomes for aging people living with HIV worldwide. However, in Uganda and other parts of sub-Saharan Africa, PLHIV with comorbidities like hypertension experience fragmented healthcare access, despite existing guidelines for the integration of non-communicable diseases into HIV care. We assessed knowledge, attitudes, and practices of PLHIV regarding hypertension care, and their perceptions of integrated hypertension-HIV care. We used a parallel convergent-mixed methods approach to collect quantitative and qualitative data from HIV clinics in urban and peri-urban Uganda. We surveyed PLHIV with hypertension to explore their knowledge, attitudes, and practices related to HTN. We selected a sub-sample from survey participants for qualitative interviews, to explore their perceptions of hypertension care and integrated HTN and HIV services. We analyzed quantitative data using STATA 14.1 and analyzed qualitative data deductively mapping it onto the Consolidated Framework for Implementation Research. A total of 394 PLHIV (325 in Kampala and 69 in neighboring Wakiso district) were enrolled in the study. Their median age was 52 years (IQR 44–59), and 300 (76%) were female. Only 32% of the participants correctly identified the normal range for systolic blood pressure (BP) (80–140 mmHg) and 24% diastolic BP (60–90 mmHg). Although 87% of the participants recognized that hypertension was treatable, only 62% knew that the treatment was lifelong. Barriers identified through interviews included fragmented care delivery, frequent hypertension medication shortages, interruptions due to side effects, high out of pocket costs of hypertension drugs, use

of Health (https://www.nhlbi.nih.gov) funded this study (UG3HL154501 awarded to authors FCS and CTL). The funders did not influence the study design, data collection and analysis, decision to publish, or the preparation of the manuscript.

**Competing interests:** The authors have declared that no competing interests exist.

of herbal remedies, and PLHIV discontinuing medication upon feeling better. Integrating chronic care for co-morbidities like hypertension in HIV clinics in Uganda offers an opportunity to address key barriers, including knowledge gaps, inconsistent medication access, and fragmented care delivery. The findings of this formative assessment informed the development of strategies to integrate hypertension-HIV care in Uganda.

## Introduction

Chronic HIV infection is known to increase the risk of cardiovascular diseases (CVD) [1], such as heart failure [2], stroke [3], and myocardial infarction [4]. Sub-Saharan Africa is disproportionately affected by HIV-related CVD, accounting for half of the global burden of disability-adjusted life-years (DALYs) lost. In some parts of Africa, the population-attributable fraction risk for HIV-associated CVD can be as high as 15%. Hypertension (HTN) is the leading risk factor for CVD in Africa [5], contributing 43% of the population-attributable risk for ischemic heart disease [6] and 15% for stroke [7].

Hypertension is a major cause of CVD disability-adjusted life years (DALYs) in Uganda, accounting for four times more cases compared to the high concentration of low-density lipoprotein cholesterol [8]. Uganda is undergoing an epidemiological transition characterized by a decline in infectious diseases coupled with an increase in non-communicable diseases (NCDs) [5]. The advent of antiretroviral therapy has enabled PLHIV to live longer, however, this has also seen a higher prevalence of age-related co-morbidities. Older PLHIV experience increased co-morbidities like hypertension having spent several years on ART [6,7]. This highlights the urgent need to strengthen health systems to address the growing burden of NCDs. The 2020 Uganda Ministry of Health (MoH) guidelines recommend integration of HIV and non-communicable disease (NCD) care, so that PLHIV receive both HIV and hypertension care within a single clinical setting, administered by the same healthcare provider, and scheduled on the same day. This integrated approach provides PLHIV-centered care, reduces fragmentation and duplication of services [9], and is more efficient than separate programs. However, to improve clinical outcomes (such as blood pressure control), existing barriers to HTN and HIV integration must be addressed. Studies in Uganda and other LMICs have identified barriers to effective HTN management, such as limited knowledge [10,11], inconsistent BP measurement, and inadequate access to medications [12]. However, there is limited information on how stakeholders' knowledge, perceptions, and experiences have been incorporated into the design of feasible and scalable implementation strategies to address these barriers and leverage facilitators to optimize integrated HTN- HIV services in HIV clinical settings. In a recent pilot study, we implemented an integrated HTN-HIV care model at a large HIV clinic in Uganda. In this study that trained healthcare providers and provided a regular supply of evidence-based anti-HTN medications to over 1000 PLHIV with HTN and HIV, up to 75.2% of PLHIV achieved HTN control at six months [13].

This work builds on the pilot study [14] to scale HTN integration to a larger, more diverse group of HIV clinics in Kampala and Wakiso districts, encompassing government-run and private not-for-profit (PNFP) facilities. This manuscript reports on the pre-implementation work aimed at identifying factors that could guide the integration of HTN services into routine HIV care by addressing existing gaps.

We utilized the findings from this formative work to create a PLHIV-centered multi-component implementation strategy to improve uptake and adherence to evidence-based BP treatments, which are contextually adapted to Ugandan HIV clinics. The strategy is being tested in a stepped-wedge cluster-randomized trial at sixteen HIV clinics in Kampala and Wakiso districts. This paper focuses on PLHIV's knowledge, attitudes, practices, and perceptions regarding HTN and integrated HTN-HIV care. The integrated model presented in this manuscript facilitates the provision of hypertension and HIV services within a "one–stop shop" framework, allowing PLHIV to access both services during a single visit. In this model, a single provider attends to PLHIV for both conditions, and all necessary medications are dispensed at a single point, other services include blood pressure measurement for screening and subsequent monitoring of hypertension control, counseling on the clinical consequences of hypertension, need for treatment (including lifestyle modification and medication) and monitoring for and mitigating any side-effects of medication.

The "Strengthening Blood Pressure Care and Treatment Cascade Metrics Among PLHIV in Peri-Urban and Urban Areas - Implementation Strategies to Save Lives (PULESA-Uganda)" study is an implementation trial aimed at improving hypertension care metrics among PLHIV in a sustainable way. The study is one of the six projects under the **H**eart, **L**ung, and **B**lood Co-morbiditie**S IM**plementation Models in **P**eople **L**iving with HIV, funded by the United States National Heart, Lung and Blood Institute. These six projects are being implemented in Uganda, Botswana, Zambia, Mozambique, Nigeria, and South Africa. [15]. Before conducting the trial, we carried out a formative study to understand the context, needs, attitudes and practices of PLHIV with hypertension who attend HIV clinics in Kampala and Wakiso districts.

The study assessed PLHIV's knowledge and attitudes toward hypertension as a health condition and hypertension management. It also identified key barriers to the integration of care. The findings from this formative study informed the development of a more contextually appropriate intervention.

### Guiding framework: The consolidated framework for implementation research

For the formative assessment, we utilized the 2009 version of the Consolidated Framework for Implementation Research (CFIR) [10], a widely recognized framework designed to identify determinants of implementation. It is important to note that this study was designed and conducted prior to the publication of the updated 2022 version of CFIR. The primary objective was to assess PLHIV's knowledge, attitudes and practices related to hypertension care alongside their perceptions of an integrated HTN_HIV care model. The CFIR constructs incorporated into the study included characteristics of individuals, patients' needs and resources, relative advantage and adaptability.

## Methods

### Study design

We conducted a parallel convergent mixed-methods study [11] that involved concurrently collecting and analyzing quantitative and qualitative data. This approach aimed to enhance the interpretability of data on PLHIV's knowledge, attitudes, practices, and perceptions of HTN and its integration into HIV care. We aimed to comprehensively understand PLHIV's knowledge, attitudes, practices, and perceptions toward HTN in the context of integrated HTN-HIV care.

### Study setting

Uganda's seven-tier healthcare system comprises national and regional referral hospitals, district hospitals, health centers (IV, III, II), and village health teams. Health center IVs offer both in-patient and outpatient services managed by medical

doctors, whereas health center IIIs offer outpatient services overseen by clinical officers and nurses. For our study, we surveyed PLHIV with HTN who were seeking HIV care at three hospitals (one public national referral hospital, two PNFP hospitals), three health center IVs, and four health center III in Kampala and Wakiso districts (**Table 1**).

## Study participants

The study involved adults who had both HTN and HIV and were receiving HIV care at selected clinics. Eligibility criteria included being 18 years or older, receiving HIV care at one of the ten participating clinics, having a confirmed hypertension diagnosis (BP ≥ 140/90 on two or more occasions in the past 12 months or on anti-hypertensive medication), and providing informed consent. We excluded individuals with cognitive impairment from the study.

We estimated the number of PLHIV with hypertension at each clinic by applying a hypertension prevalence rate of 20.9% [12]. We obtained the number of PLHIV with hypertension at each clinic and the overall total. We then calculated a proportion for each clinic by dividing the estimated number of PLHIV with hypertension at each clinic by the overall total and applied this proportion to the calculated sample size. This approach ensured that PLHIV were recruited from each clinic in equal proportions. However, the target sample size was not reached in clinics where hypertension screening was not conducted, or documentation was lacking. Using consecutive sampling, we enrolled eligible PLHIV through chart reviews and healthcare provider referrals at the participating clinics. For the qualitative component, we purposively selected a sub-sample of four PLHIV per facility (n = 40), using maximum variation to identify individuals by age, sex, and duration of HTN (newly diagnosed versus those with a known diagnosis). Newly diagnosed PLHIV were those diagnosed with HTN within the preceding six months, while previously diagnosed PLHIV had been diagnosed for six months or more. There were no refusals from PLHIV to participate in qualitative interviews.

## Data collection

Our study employed a parallel convergent design therefore data collection for both quantitative and qualitative data was conducted concurrently between 15th June and 31st December 2021. Quantitative data was collected through a survey questionnaire while interviews were conducted via a semi-structured interview guide.

**Table 1. Sites where the formative component of the PULESA Uganda study was conducted.**

| Clinic site | Clinic type | Clinic size* | Location | No. PLHIV | Estimated PLHIV with HTN | KAP Survey | IDI |
|---|---|---|---|---|---|---|---|
| **Kampala** | | | | | | | |
| Kisenyi HC IV | Public | Large | Urban | 11972 | 2502 | 118 | 4 |
| Kawaala HCIV | Public | Large | Urban | 8814 | 1842 | 86 | 4 |
| Komamboga HC III | Public | Large | Urban | 4732 | 989 | 30 | 4 |
| St. Francis Hospital | PNFP | Large | Urban | 7911 | 1653 | 78 | 4 |
| Butabika Hospital | Public | Small | Urban | 1310 | 274 | 13 | 4 |
| **Wakiso** | | | | | | | |
| Kisubi Hospital | PNFP | Large | Peri-urban | 2221 | 464 | 22 | 4 |
| Nsangi HCIII | Public | Small | Peri-urban | 1865 | 390 | 18 | 4 |
| Kawanda HC III | Public | Small | Peri-urban | 830 | 173 | 8 | 4 |
| Nakawuka HCIII | Public | Small | Peri-urban | 852 | 178 | 7 | 4 |
| Kira HCIV | Public | Small | Peri-urban | 1120 | 234 | 14 | 4 |
| **Total** | | | | **42021** | **8782** | **394** | **40** |

IDI; In-depth interview HC; Health Centre, KAP; Knowledge, attitudes and practice, PNFP; Private not for profit

*Large HIV clinics were defined as clinics providing HIV care to >2000 PLHIV and small clinics were clinics that provided HIV care to 400–2000 PLHIVs at the time of the study.

**Quantitative data.** We conducted a survey and used a questionnaire (S1 Text) to gather information on different aspects of hypertension (HTN). We included questions about PLHIV's socio-demographic characteristics, lifestyle factors (such as tobacco use and excessive alcohol consumption), medical and family history, as well as their knowledge, attitudes, and practices (KAP) regarding HTN. The survey comprised 28 questions assessing PLHIV's knowledge, 12 questions on attitudes, and 10 questions on practices related to HTN. Topics covered included screening, diagnosis, risk factors, treatment, complications, and lifestyle modifications.

Trained Research assistants enrolled eligible PLHIV and administered a pre-tested questionnaire sequentially until each health facility's targeted sample size was achieved (Table 1). The survey was conducted in Luganda, a commonly spoken language in central Uganda, with a few PLHIV interviewed in English. The questionnaire was pretested at a large tertiary clinic in Kampala, which was not part of the study clinics.

**Qualitative data.** Qualitative data exploring PLHIV's perceptions of hypertension and integrated care were gathered through in-depth interviews. A semi-structured interview guide (S2 Text), informed by the 2009 version of CFIR, was developed to ensure a systematic exploration of key constructs. Trained research assistants – male and female university graduates fluent in Luganda and English, with extensive experience in qualitative research conducted the interviews. A total of forty (n = 40) face -to-face interviews were carried out in private spaces within clinics or health facilities, each lasting between 45 and 60 minutes. All interviews were audio-recorded, transcribed verbatim, and translated into English to facilitate analysis. Data collection continued until thematic saturation was reached, which was determined by the 40th interview [16]. To enhance the rigor of the study, the interview guide was pre-tested at a non-participating HIV clinic in Kampala, ensuring clarity and appropriateness of the questions.

### Ethical considerations

The research study received approval from the Makerere University School of Medicine Research and Ethics Committee (SOMREC) (Mak-SOMREC-2021–58) and was registered with the Uganda National Council for Science and Technology (UNCST) (SS808ES). Administrative clearance was obtained from the MoH, Kampala Capital City Authority, and Wakiso District Local Government. Before participating in the study, all PLHIV provided written informed consent.

### Data analysis

We analyzed quantitative and qualitative data separately (Table 2) and combined them to understand PLHIV's knowledge, attitudes, and practices regarding HTN, as well as their views on an integrated HIV-HTN care model.

### Quantitative data analysis

The survey questions were developed in a REDCap database [14,17] and we used STATA version 14.1 for quantitative data analysis. For statistical analysis, categorical data were summarized using proportions and frequencies. For normally distributed continuous data, means and standard deviations were used, while median and interquartile range were used for skewed continuous data. Composite scores for each KAP domain were calculated, and district comparisons were performed using chi-squared tests (categorical variables) and Mann-Whitney tests (continuous variables).

**Table 2. Data Sources and approaches for data collection and analysis in the formative study.**

| Study objective | Method | Study Population | Data Analysis |
|---|---|---|---|
| Assess PLHIV's knowledge, attitudes and practices about HTN, HTN management, and control in HIV clinics in Kampala and Wakiso districts. | Quantitative KAP Survey questionnaire | PLHIV with HTN | descriptive analysis |
| Assess PLHIV's perceptions of current HTN management and control practices in HIV clinics in Kampala and Wakiso Districts. | Qualitative Semi-structured interviews | PLHIV with HTN | Deductively based on CFIR constructs. |

HTN- HTN, KAP- Knowledge Attitudes and Practices, PLHIV- PLHIV.

## Qualitative data analysis

We used NVivo 20 [18,19] to manage the qualitative dataset. In our analysis, we followed a team-based approach to identify emerging themes and develop a preliminary codebook inductively. We held a series of meetings to discuss emerging themes through consensus building. We then utilized a deductive approach to align the emerging themes from inductive coding and mapped them onto the CFIR domains and constructs. The larger research team discussed the preliminary themes and triangulated responses from semi-structured interviews with KAP survey data to reach a consensus on the final deductive themes.

We followed the established criteria and standards for reporting qualitative research (S1 COREQ Checklist) and maintained an audit trail of the coding process to ensure credibility and trustworthiness [20,21].

## Data integration

We used a parallel convergent design for this study. Data were integrated [11] in two ways. We collected quantitative and qualitative data simultaneously, a process known as mixed methods integration. Secondly, we utilized the connecting method, linking quantitative data to qualitative data through sampling. The qualitative sample was a subset of the quantitative sample.

We combined both approaches better understand HTN care practices in HIV clinics, and PLHIV's perceptions of HTN-HIV care integration. **Fig 1** provides a visual representation of the mixed methods design used in this study.

# Results

The results of this study are structured around the key domains and constructs of the CFIR, which guided our formative assessment. These domains—characteristics of individuals, patients' needs and resources, relative advantage, and adaptability—provide a comprehensive framework for understanding the factors influencing the integration of HTN care in HIV clinics.

## Characteristics of study participants

A total of 537 PLHIV with HTN were identified for the study. Of these, 439 (81.7%) were successfully contacted, and 394 (73.4%) participated in the survey (Fig 2**). The median age of PLHIV was 52 years (IQR 44–59), with a higher proportion of females (76.1%, n = 300). Most (51.3%, n = 202) had only completed primary-level education participants (**Table 3**). Geographically, most PLHIV were from Kampala (n = 325), with 69 PLHIV from Wakiso.

**Knowledge and beliefs about hypertension.** The analysis of PLHIV knowledge revealed notable gaps, particularly in understanding the etiology and clinical thresholds associated with HTN. Awareness surrounding hypertension-related

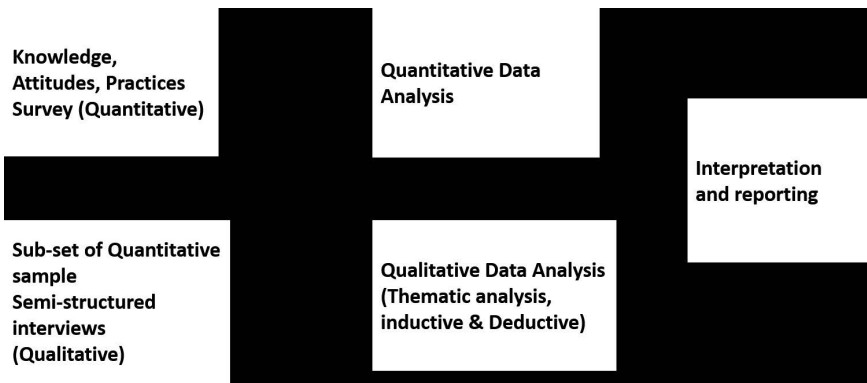

**Fig 1. Visual representation of how methods were mixed using the parallel convergent mixed methods approach.**

risk factors, diagnosis, treatment, and complications was found to be insufficient among PLHIV. The level of knowledge among PLHIV regarding clinical thresholds for diagnosing hypertension was low. Only 32% of PLHIV correctly identified the systolic BP threshold of 140 mmHg for diagnosing hypertension (HTN), while 24.1% accurately identified the diastolic BP threshold of 90 mmHg. Despite these significant gaps in diagnostic knowledge, most participants, PLHIV (86.8%) acknowledged that hypertension is treatable, and 62% understood that lifelong management is required for those diagnosed with the condition (Table 4).

In addition to quantitative data, qualitative interviews provided further insight into PLHIV beliefs about hypertension. PLHIV frequently attributed hypertension to stress induced by life challenges and identified physical symptoms, such as increased heart rate, headaches, weakness, and insomnia, as indicative of the condition. One PLHIV articulated their perception of hypertension as follows:

*"….. it causes constant heartbeat, headache, feeling weak. I believe your BP rises when dealing with life's problems. You may get a headache or sometimes feel dizzy"* **(Male, 56 years).** Other PLHIV also described symptoms such as palpitations and difficulty in breathing as associated with hypertension.

**Attitudes toward hypertension screening, treatment and monitoring.** Most PLHIV strongly supported regular blood pressure (BP) checkups and recognized the importance of lifestyle modifications in preventing hypertension. However, misconceptions about HTN were prevalent. A significant proportion (90%) believed that hypertension always presents with symptoms, despite its frequently asymptomatic nature. Additionally, two thirds (66%) of PLHIV expressed belief in the efficacy of herbal treatments, and 42% considered it acceptable to discontinue antihypertensive medications in the absence of symptoms (Fig 3).

During in-depth interviews, PLHIV acknowledged the critical need for routine BP screenings to monitor their condition and prevent complications. One PLHIV highlighted the dangers of undiagnosed and uncontrolled hypertension.

*"You may get a stroke: you may have uncontrolled (high) BP, which will affect your eyes…you may lose a limb through amputation, so it is important for us to be screened because it is difficult to know you have it until you are screened."* **(Female, 49 years)**

Despite this awareness, findings indicate gaps in comprehensive PLHIV's education regarding hypertension management and the necessity of continuous treatment adherence. While PLHIV expressed positive attitudes toward BP

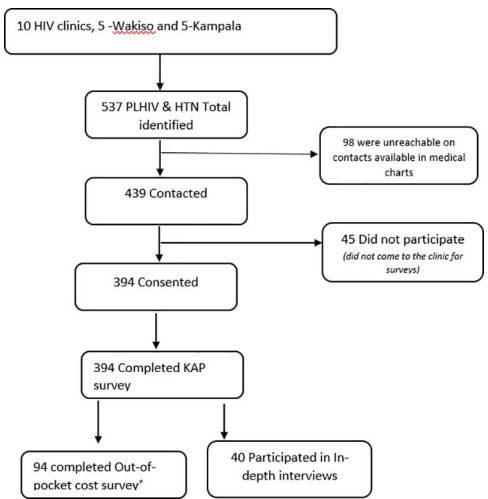

**Fig 2. Study Participant Flow Diagram.**

**Table 3.** Demographic characteristics of PLHIV participating in the formative study at selected HIV clinics in Kampala and Wakiso districts.

| Variable | Overall(n=394) | Kampala(n=325) | Wakiso (n=69) | p-value |
|---|---|---|---|---|
| Female sex | 300 (76.1) | 246 (75.7) | 54 (78.3) | 0.649 |
| Median age (IQR)ᵃ | 52 (44, 59) | 52 (45, 59) | 53 (42, 60) | 0.751 |
| Distance to clinic(kms) (KmKmsKm (KM) | 8 (4, 15) | 8 (5, 16) | 5 (3, 10) | **0.003** |
| Education level | | | | |
| No formal education | 37 (9.4) | 31 (9.5) | 6 (8.7) | 0.556 |
| Primary | 202 (51.3) | 168 (51.7) | 34 (49.3) | |
| Secondary | 121 (30.7) | 101 (31.1) | 20 (29) | |
| Post-secondary | 34 (8.6) | 25 (7.7) | 9 (13) | |
| Marital status | | | | |
| Married/Cohabiting | 130 (33) | 105 (32.3) | 25 (36.2) | 0.167 |
| Single/widowed | 141 (35.8) | 123 (37.8) | 18 (26.1) | |
| Divorced | 123 (31.2) | 97 (29.9) | 26 (37.7) | |
| Occupation | | | | |
| Formally employed | 60 (15.2) | 51 (15.7) | 9 (13) | 0.356 |
| Self-employed | 172 (43.7) | 148 (45.5) | 24 (34.8) | |
| Peasant farmer | 29 (7.4) | 23 (7.1) | 6 (8.7) | |
| Unemployed | 118 (30) | 91 (28) | 27 (39.1) | |
| Other | 15 (3.8) | 12 (3.7) | 2 (4.4) | |

ᵃWilcoxon rank sum test, Categorical variables – Chi-square test.

monitoring and lifestyle modifications, concerns about polypharmacy and potential long-term effects of antihypertensive medication were prevalent. One PLHIV shared concerns about excessive medication use:

*"However, we were also told that having a lot of medication in one's body could lead to cancer. I got scared, and now I do not take the [anti-HTN] pill consistently. I may take the medication on one day and skip two days."* **(Female, 58 years)**

Adverse effects of medication also emerged as a major factor hindering adherence. Many PLHIV described experiencing side effects such as dizziness and fatigue, which deterred consistent medication use. One PLHIV noted:

*"When you take medication for high BP, you become very weak. The tablets [antihypertensive medicines] make you weak; you feel dizzy. They cause dizziness, you feel weak, and your whole body gets weak."* **(Female, 42 years)**

These findings suggest that the perceived and actual side effects of antihypertensive medications contribute to inconsistent adherence, potentially compromising effective hypertension management. Fear of long-term medication side effects, such as cancer further compounded adherence challenges, highlighting the need for targeted interventions to address misinformation and enhance PLHIV support.

## Outer setting

**Patient needs and resources.** The study examined PLHIV needs and the extent to which barriers and facilitators to meeting those needs were recognized and prioritized. Findings indicate that that HTN services within HIV clinics were not prioritized, contributing to significant gaps in care. Routine blood pressure screening was infrequent, limiting early detection and management of hypertension, while inadequate access to anti-hypertensive medications further impeded PLHIV care.

Only 44% of PLHIV, when asked about the last time they had their blood pressure checked, reported having had it checked within the last seven days, while 24% were screened between eight and thirty days prior. Notably, PLHIV from urban clinics were more likely to receive hypertension treatment compared to those from peri-urban clinics (64.3% vs. 47.8%, p=0.003) (Table 5).

**Table 4. PLHIV knowledge about HTN diagnosis, treatment, monitoring and complications.**

| Variable | Overall (n = 394) | Kampala (n = 325) | Wakiso (n = 69) | p value |
|---|---|---|---|---|
| **Threshold systolic BP** | | | | |
| 80 – 140 mmHg | 126 (32) | 110 (33.9) | 16 (23.2) | 0.085 |
| Others | 268 (68) | 215 (66.1) | 53 (76.8) | |
| **Threshold normal diastolic BP** | | | | |
| 60 -90 mmHg | 95 (24.1) | 81 (24.9) | 14 (20.3) | 0.414 |
| Others | 299 (75.9) | 244 (75.1) | 55 (79.7) | |
| **Can high BP be treated?** | | | | |
| Yes | 342 (86.8) | 282 (86.8) | 60 (87) | 0.574 |
| No No | 13 (3.3) | 12(3.7) | 1 (1.4) | |
| I don't know | 39 (9.9) | 31 (9.5) | 8 (11.6) | |
| **Duration of taking anti-HTN medicines** (n = 342) | | | | |
| For life | 212 (62) | 173 (61.3) | 39 (65) | 0.636 |
| Does not know | 96 (28.1) | 82 (29.1) | 14 (23.3) | |
| Only when BP is high/abnormal | 34 (9.9) | 27 (9.6) | 7 (11.7) | |
| **HTN damages body organs.** | | | | |
| Yes | 344 (87.3) | 282 (86.8) | 62 (89.9) | |
| No | 21 (5.3) | 16 (4.9) | 5 (7.2) | |
| Does not know | 29 (7.4) | 27 (8.3) | 2 (2.9) | |
| **Body Organs damaged by HTN** | | | | |
| Brain | 156 (39.6) | 125 (38.5) | 31 (44.9) | 0.319 |
| Eye | 83 (21.1) | 62 (19.1) | 21 (30.4) | **0.036** |
| Heart | 132 (33.5) | 105 (32.3) | 27 (39.1) | 0.275 |
| Kidneys | 27 (6.9) | 23 (7.1) | 4 (5.8) | 0.702 |
| Others | 5 (1.3) | 4 (1.2) | 1 (1.4) | 0.883 |

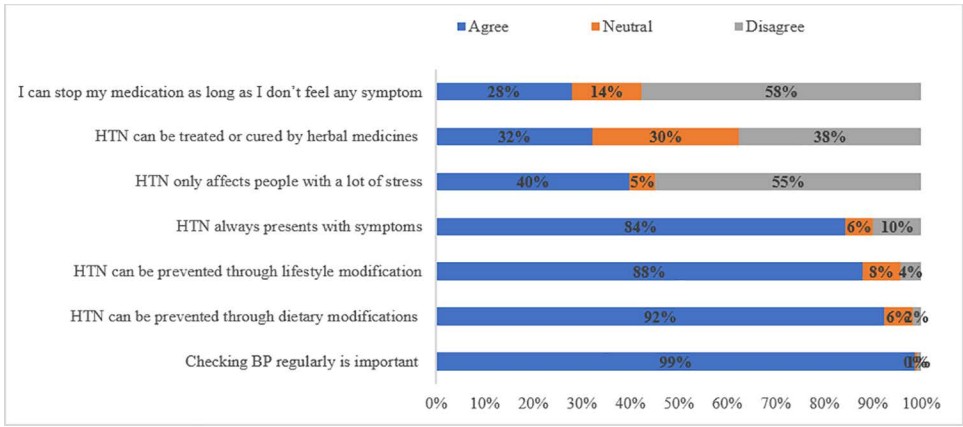

**Fig 3. Attitudes towards HTN diagnosis, risk factors and complications among PLHIV with HTN receiving HIV care in Kampala and Wakiso districts.**

Qualitative findings revealed PLHIV challenges in accessing hypertension care, primarily due to the inadequacy of services provided at HIV clinics. Routine BP screening, medication management, and clinical monitoring were often lacking, leading to fragmented care experiences. One PLHIV described this issue:

**Table 5. PLHIV's practices regarding HTN diagnosis, treatment, monitoring and screening.**

| | Overall(n=394) | Kampala (n=325) | Wakiso (n=69) | p value |
|---|---|---|---|---|
| **Last BP measurement** | | | | |
| Last 7 days | 173 (43.9) | 152 (46.8) | 21 (30.4) | 0.089 |
| 8- 30 days | 96 (24.4) | 74 (22.8) | 22 (31.9) | |
| 31- 120 days | 58 (14.7) | 45 (13.8) | 13 (18.8) | |
| 121 days | 67 (17) | 54 (16.6) | 13 (18.8) | |
| **Frequency of taking anti-HTN medications** | | | | |
| Daily | 242 (61.4) | 209 (64.3) | 33 (47.8) | 0.003 |
| Weekly | 5 (1.3) | 4 (1.2) | 1 (1.4) | |
| When I feel unwell | 70 (17.7) | 47 (14.5) | 23 (33.3) | |
| Others | 77 (19.5) | 65 (20) | 12 (17.4) | |
| **Assessed for HTN-induced complications** | | | | |
| I have never been assessed | 317 (80.5) | 263 (80.9) | 54 (78.3) | 0.061 |
| Monthly | 10 (2.5) | 9 (2.8) | 1 (1.4) | |
| Every three months | 12 (3.1) | 11 (3.4) | 1 (1.4) | |
| Every six months | 7 (1.8) | 7 (2.1) | 0 | |
| Every 12 months | 40 (10.1) | 27 (8.3) | 13 (18.8) | |
| Does not know | 8 (2) | 8 (2.5) | 0 (0) | |

*"They [healthcare providers] do not measure our BP. I have never received any serious advice concerning HTN."* **(Female, 52 years)**

This indicates that systematic monitoring and management of hypertension were not routinely integrated into HIV care settings, potentially compromising PLHIV outcomes. PLHIV with hypertension were frequently referred to other service points either within or outside the facility, further exacerbating fragmentation. One PLHIV explained:

*"There are no efforts to provide treatment within this HIV clinic apart from advising you to seek treatment for HTN privately."* **(Female, 54 years)**

The referral process increased the burden on PLHIV, leading to delays in treatment and inadequate continuity of care.

A major barrier to hypertension management was the unavailability of antihypertensive medications at public healthcare facilities, forcing PLHIV to seek medications privately. However, private clinics often charged prices beyond PLHIV's financial means, as described by one PLHIV:

*"At this clinic, we do not receive any treatment for HTN. Once your BP is measured, you are often advised to go and buy antihypertensive medications privately, but the medicines are expensive."* **(Female, 35 years)**

The high cost of antihypertensive medications significantly impeded adherence, leading PLHIV to ration medications. One PLHIV stated:

*"I may skip two days without taking [antihypertensive] medication, and when I can afford it, I resume."* **(Female, 58 years)**

Medication rationing compromises treatment efficacy and exacerbates hypertension progression, increasing the risk of adverse long-term health outcomes.

PLHIV reported a lack of BP screening services and counseling on hypertension management, primarily due to insufficient staffing at HIV clinics. Many advocated for improvements including better access to BP machines and integrated care services. One PLHIV emphasized:

*"I request that more staff manage HTN, more effort [towards] screening clients for HTN. I would also like that whenever I come to the clinic, my BP is checked and I am given treatment, have all [HIV, HTN] services provided at once, and all medicines given at once."* (Female, 49 years)

The findings highlight the need for a streamlined and comprehensive approach to managing both HIV and hypertension to improve PLHIV access and promote better health outcomes.

**Intervention characteristics**

**Perceived relative advantage and adaptability.** Our findings highlight two key constructs within the CFIR domain of intervention characteristics; relative advantage and adaptability.

PLHIV perceived the integration of hypertension care within HIV services as a more beneficial alternative to the current fragmented and disjointed care models. They also expressed confidence that hypertension management could be successfully incorporated into existing HIV infrastructure, offering both logistical and financial benefits.

PLHIV identified several advantages of integrating HTN and HIV services, particularly in mitigating barriers related to fragmented care and the high cost of antihypertensive medications. A primary benefit emphasized was the convenience of receiving both treatments within a single visit, which would reduce logistical challenges and financial strain. As one PLHIV stated:

*"For a hypertensive person, it helps when your BP is measured, and you get to know your status whether it is controlled or not. The other issue is access to medicine: in most cases we come here, and we don't find the [antihypertensive] medicines."* **(Female, 26 years)**

This sentiment underscores dissatisfaction with the current model, where the unavailability of antihypertensive medications at HIV clinics disrupts continuity of care. PLHIV highlighted that integrating HTN services within HIV care would minimize the time spent seeking treatment, improve medication access, and lower financial burdens. The ability to simultaneously obtain ART and antihypertensive medications was particularly valued. One PLHIV explained:

*"I would be very happy if I can get both ART and antihypertensive medicines at the same time. This would mean I only spend one day to get everything I need. This would be best for me."* **(Female, 59 years)**

This preference for a consolidated care model suggests that integrating HTN management into HIV services could significantly enhance PLHIV adherence and overall health outcomes. In addition to convenience, PLHIV highlighted the potential financial benefits of integrating HTN and HIV care. They emphasized that consolidating clinic visits would reduce transportation costs and other expenses associated with seeking care at multiple locations. One PLHIV noted:

*"The benefit of [HTN-HIV] is that one does not have to travel several times to the facility. For example, if I come on my clinic day, my BP is measured, and treatment is received, it would greatly benefit me."* **(Female, 49 years)**

This finding suggests that integrating services could improve financial accessibility, particularly for individuals with limited resources who struggle with transportation costs and clinic fees.

PLHIV also emphasized the feasibility of integrating HTN care into existing HIV service delivery models, particularly by aligning hypertension management with established HIV care structures such as differentiated service delivery models – multi-month dispensing and community based services. The majority expressed a preference for synchronized refills of antihypertensive medications alongside their HIV treatment, as illustrated by one PLHIV's recommendation:

*"Let the treatment [antihypertensive medication] be given to us at once: a package for 3 months – [high blood] pressure [medicines for] 3 months, diabetes [medicines for] 3 months, HIV [medicines for] 3 months – and you know that after the 3 months, I will go back to the hospital."* **(Male, 54 years)**

The preference for synchronized medication refills highlights the need for a streamlines, efficient care process that reduces the frequency of clinic visits while ensuring uninterrupted treatment adherence. Furthermore, PLHIV emphasized the importance of addressing transportation challenges and making healthcare services more accessible. One PLHIV explained:

*"Some of us clients don't have transport to come to the clinic. We shall benefit in the way that HIV clients – they have thought about us – to bring us treatment for other conditions nearby, rather than coming today for one condition and another day for another condition. So from here, we shall get good services."* **(Female, 38 years)**

This perspective underscores the need for integrated care models that reduce logistical barriers, particularly for individuals managing multiple chronic conditions. Additionally, financial constraints were identified as a significant barrier to full treatment access, as illustrated by one PLHIV's concern:

*"I think it would be good if we collectively receive treatment. Sometimes when we do not get medicines, we find that we do not have money or can only make a partial payment, and then we find that we have not received full treatment. That is the only problem,"* **(Female, 46 years)**

## Discussion

This study assessed PLHIV's knowledge, attitudes, and practices regarding hypertension and hypertension management and identified key barriers affecting the integration of HTN and HIV services. It highlights the need for PLHIV-level interventions to optimize HTN management within HIV clinics, as PLHIV demonstrated significant gaps in understanding HTN, including its causes, treatment, and self-management. These gaps hindered PLHIV engagement and adherence, underscoring the importance of education and awareness in successful integrated care.

Building on prior research, the study suggests leveraging HIV infrastructure for HTN education, a strategy shown to improve adherence and awareness in other African contexts, such as Kenya [22]. In sub-Saharan Africa, HIV care infrastructure has often incorporated robust health education components, often facilitated by peer support programs that offer both education and emotional support [23]. This study builds upon this evidence, showing that leveraging the existing HIV infrastructure for HTN education could be a promising strategy; training HIV care providers in HTN management is crucial for improving PLHIV education and self-management practices, especially among older PLHIV, who often face compounded challenges with multiple chronic conditions. Previous studies emphasize the pivotal role of provider knowledge and communication skills in chronic disease management, influencing both adherence and clinical outcomes. Our study corroborates previous research on aging with HIV and underscores the need to leverage HIV programs to provide services to older adults living with HIV who are most susceptible to co-morbidities specifically in low and middle income countries like Uganda [24]. The study also identified inconsistent access to affordable antihypertensive medications as a major barrier. While public health facilities offer free medications, stock-outs force PLHIV to purchase them privately, presenting a significant challenge in low-resource settings [25]. Addressing medication affordability and availability, particularly for older PLHIV who may be more financially vulnerable, is essential for effective integrated care. Research indicates that older adults in low-resource settings often experience difficulties accessing comprehensive care due to age-related vulnerabilities, including polypharmacy and the burden of managing multiple health conditions [26,7].

Financial constraints were also identified as a barrier, particularly for older PLHIV who struggle with the costs of healthcare, including clinic visits, travel, and medications. The study suggests a model that reduces economic burdens on PLHIV, particularly older adults managing multiple chronic conditions. Research in Uganda and low-resource settings has shown similar cost-related challenges in managing chronic diseases such as diabetes and rheumatic heart disease [27].

Aligning hypertension (HTN) care with existing differentiated service delivery (DSD) models for HIV care could reduce clinic visit frequency and enhance service utilization, particularly for older PLHIV. Effective integration is crucial, as DSD models—widely implemented across Africa for clinically stable ART clients—have proven beneficial in promoting client-centered care by adapting service delivery to meet individual needs (30). During the COVID-19 pandemic, community-based ART distribution models were strengthened to maintain continuity of care outside of health facilities [28]. PLHIV in our study emphasized the need for HTN services to be adapted similarly, allowing for synchronized medical refills and reduced visit frequency. Such alignment could enhance service utilization, reduce the risk of treatment interruptions, and improve health outcomes for PLHIV managing both HIV and HTN.

In summary, integrating HTN and HIV care emerged as a promising model, offering benefits in accessibility, cost savings, and convenience for PLHIV. The adaptability of this model within the existing HIV care infrastructure highlights potential synergies in treatment delivery that could address both logistical and financial barriers to care. Integrating HTN

care with existing DSD models, ensuring a consistent medication supply, and enhancing PLHIV education can improve medication adherence, reduce transportation costs, and address the broader challenges of chronic disease management in under-resourced health systems. Given the growing burden of non-communicable diseases among aging PLHIV in Uganda, implementing integrated service models that cater to their unique healthcare needs is increasingly imperative. Future research should explore the long-term impact of integration on clinical outcomes and client retention, particularly among older PLHIV who face additional socioeconomic and health-related vulnerabilities.

### Strengths of this study

Our study employed both quantitative and qualitative methods to examine the implementation determinants of integrated hypertension and HIV care from the perspective of PLHIV. The robust sample size enabled precise estimates and meaningful comparisons across the two districts. Additionally, the inclusion of a diverse range of clinics—spanning small and large private not-for-profit and public facilities—enhances the generalizability of our findings compared to previous studies. These findings provide critical insights into the gaps and implementation challenges that must be addressed to integrate HTN care into HIV clinic settings effectively.

### Limitations

The study had some limitations. The formative assessment was only carried out in urban and peri-urban areas of Kampala and Wakiso districts. These facilities may not necessarily represent the characteristics of rural communities that could be more limited in resources. However, other interventions primarily conducted in urban and peri-urban settings have been successfully implemented on a broader scale [29,30].

### Conclusion

Integrating hypertension services into HIV clinics in Uganda presents a critical opportunity to enhance chronic disease management by addressing key barriers such as knowledge gaps, inconsistent medication access, and fragmented care delivery. Effective integration will require coordinated efforts among policymakers, healthcare providers, and community stakeholders to develop a cohesive, PLHIV-centered models that optimize service delivery that supports dual management of HIV and HTN within resource-constrained settings. Additionally, incorporating structural health education on comorbidities into routine HIV care is essential for improving PLHIV awareness, engagement, and adherence to hypertension management. The findings from this study provide evidence-based guidance for designing targeted integration strategies that address PLHIV-specific challenges including financial constraints and misalignment of services, ultimately strengthening the healthcare system's capacity to manage the dual burden of HIV and hypertension. Furthermore, these insights underscore the importance of incorporating health education on comorbidities into routine HIV care to foster PLHIV's understanding and engagement. This study's findings informed targeted strategies to integrate HTN care into HIV settings, focusing on PLHIV-specific barriers such as knowledge deficits, financial constraints, and service misalignment.

### Contributions to the literature

- This study identified the importance of PLHIV level implementation determinants to deliver PLHIV centered care through integrated HTN and HIV services in Uganda

- This study illustrates the need to identify and leverage existing HIV infrastructure and resources as a pathway to optimizing non-communicable disease care for an aging PLHIV population in resource limited settings like Uganda

- Findings identified through this study have been utilized to develop a multi-component intervention to optimize integration of HTN care into the HIV program. The intervention is being implemented in a stepped wedge cluster randomized control trial across 16 HIV clinics in Uganda

## Supporting information

**S1 Text. KAP Survey Questionnaire.**
(DOCX)

**S2 Text. Indepth Interview Guide.**
(DOCX)

**S1 Checklist. COREQ Checklist.**
(DOCX)

## Acknowledgments

The authors would like to thank all the research participants who agreed to participate in this study. The authors acknowledge the support of the Ministry of Health in Uganda, Wakiso District Local Government and Kampala Capital City Authority, which granted access for researchers to engage with participants receiving HIV and HTN care from health facilities within their jurisdiction. The authors acknowledge the Makerere University Behavioral and Social Science Research (Mak-BSSR) programme which is funded by the National Institutes for Health (NIH), National Institutes of Health on Alcohol Abuse and Alcoholism (NIAAA), National Institute of Mental Health (NIMH), and Fogarty International Center (FIC), Grant number: D43TW011304 (Kamya, Camlin, and Katahoire) under which Florence Ayebare (Corresponding author) is a trainee.

## Author contributions

**Conceptualization:** Fred C. Semitala, Gerald N. Mutungi, Isaac Ssinabulya, James Kayima, Martin Muddu, Donna Spiegelman, Jeremy I. Schwartz, Chris T. Longenecker, Anne R. Katahoire.

**Data curation:** Florence Ayebare, John Baptist Kiggundu, Christine Kiwala, Joel Senfuma, Anne R. Katahoire.

**Formal analysis:** Florence Ayebare, John Baptist Kiggundu, Christine Kiwala, Joel Senfuma, Anne R. Katahoire.

**Funding acquisition:** Fred C. Semitala, Chris T. Longenecker.

**Investigation:** Fred C. Semitala, Chris T. Longenecker.

**Methodology:** Fred C. Semitala, John Baptist Kiggundu.

**Project administration:** John Baptist Kiggundu.

**Resources:** John Baptist Kiggundu.

**Software:** Florence Ayebare, John Baptist Kiggundu.

**Supervision:** Florence Ayebare, John Baptist Kiggundu, Anne R. Katahoire.

**Validation:** Fred C. Semitala, Florence Ayebare, John Baptist Kiggundu, Chris T. Longenecker, Anne R. Katahoire.

**Writing – original draft:** Florence Ayebare.

**Writing – review & editing:** Fred C. Semitala, Florence Ayebare, John Baptist Kiggundu, Christine Kiwala, Joel Senfuma, Gerald N. Mutungi, Isaac Ssinabulya, James Kayima, Martin Muddu, Donna Spiegelman, Jeremy I. Schwartz, Chris T. Longenecker, Anne R. Katahoire.

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
