## [Decision Letter · Decision Letter 0]

PGPH-D-24-01877

Harnessing HIV clinics to deliver integrated hypertension care for People living with HIV in Uganda: A mixed methods study.

Dear Dr. Ayebare,

Thank you for submitting your manuscript to PLOS Global Public Health. After careful consideration, we feel that it has merit but does not fully meet PLOS Global Public Health’s publication criteria as it currently stands. Therefore, we invite you to submit a revised version of the manuscript that addresses the points raised during the review process.

EDITOR: 

Address typographic errors noticed in several sections of the documentPrior to using an abbreviation, spell it out in full. For example, first use of PLHIV in the introduction section is not spelt out

We look forward to receiving your revised manuscript.

Kind regards,

Jepchirchir Kiplagat, PhD

Academic Editor

Journal Requirements:

**Please only choose the relevant sentences from below**

1. Please clarify all sources of funding (financial or material support) for your study. List the grants (with grant number) or organizations (with url) that supported your study, including funding received from your institution. 

2. State the initials, alongside each funding source, of each author to receive each grant.

3. State what role the funders took in the study. If the funders had no role in your study, please state: “The funders had no role in study design, data collection and analysis, decision to publish, or preparation of the manuscript.”

4. If any authors received a salary from any of your funders, please state which authors and which funders.

2. In the online submission form, you indicated that "The datasets used/analyzed during the current study are available from the corresponding author upon reasonable request.". 

3. Uploaded as supplementary information.

3. Please upload a copy of Figures which you refer to in your manuscript. Or, if the figure is no longer to be included as part of the submission please remove all reference to it within the text.

Please provide separate figure files in .tif or .eps format.

4. We have noticed that you have uploaded Supporting Information files, but you have not included a list of legends. Please add a full list of legends for your Supporting Information files after the references list. 

Reviewers' comments:

Reviewer's Responses to Questions

**Comments to the Author**

1. Does this manuscript meet PLOS Global Public Health’s publication criteria? Is the manuscript technically sound, and do the data support the conclusions? The manuscript must describe methodologically and ethically rigorous research with conclusions that are appropriately drawn based on the data presented.

Reviewer #1: Yes

Reviewer #2: Partly

2. Has the statistical analysis been performed appropriately and rigorously?

Reviewer #1: Yes

Reviewer #2: Yes

3. Have the authors made all data underlying the findings in their manuscript fully available (please refer to the Data Availability Statement at the start of the manuscript PDF file)?

Reviewer #1: Yes

Reviewer #2: Yes

4. Is the manuscript presented in an intelligible fashion and written in standard English?

Reviewer #1: Yes

Reviewer #2: Yes

5. Review Comments to the Author

Reviewer #1: Overall: An interesting and very timely analysis that supports previous findings about barriers to successful implementation of hypertension programming in sSA.

What is the integration model used here at systems and program level? Please describe it in the introduction. Often we say programs are ‘integrated’ without providing necessary detail to understand the integration model. Here, please tell us what systems are integrated (unified supply chain? Does the EMR record both HTN and HIV data? Does the same cadre or health care workers screen and diagnose both diseases? Lab integration?) and what program elements are integrated in this model (coordinated pharmacy pickups for ART and HTN meds? Multimonth dispensing of both types of medications? Differentiated service delivery options for both ART and HTN meds? Same clinic space for both services? Same clinical provider for both services? Same provider gives lifetstyle/behavioral counseling? etc). Were integration models for HTN/HIV different at the public and private hospitals that you surveyed? How? How might these different models have impacted client survey responses?

Introduction:

Nice review of relevant literature. Clear explanation of how this current study fits into recent and ongoing pilot studies this group is doing with HIV/HTN integration.

Line 72-73: reference?

Line 72: please explain the difference between hypertension and high blood pressure?

Line 74-75: reference?

Line 78: “all PLHIV”: is there an age cutoff at which BP screening should start in these guidelines? Usually age 18 and older- would specify here

Line 79-80: “PLHIV who are diagnosed with HTN should receive integrated services for both HTN and HIV”. What is specifically meant by ‘integrated services’ here? Do the Ugandan guidelines specify or define the term ‘integration’ or a particular integration model(s)? Throughout this paper, the only integration model referred to is integrating NCDs into HIV care.

Line 82-83: “However, for successful combined health outcomes for HTN-HIV, it is crucial to follow these guidelines.” This sounds like an advocacy statement. Consider rephrasing to indicate that existing barriers to HTN and HIV integration may limit efficacy of HTN programs.

Line 93: It seems this work is part of the PULESA Uganda study? I would devote a paragraph in the introduction to explaining the PULESA work/findings/integration model to date, and how this current study fits in.

Line 107: would remove “for example data collection tools were developed based on the CFIR 2009” as this is already explained in lines 104-105

Line 109: “integrated model”: still unclear what this integrated model actually entails.

Line 109-112: “Following analysis, the team agreed that the following 110 constructs were relevant for the study to develop strategies to address patient level gaps to inform intervention components for patient centered care in the larger Trial.” Consider moving to Discussion as it is not relevant in the Introduction.

Line 113-114: “We used the CFIR framework to identify implementation

114 determinants to improve HTN care within the HIV program.” Consider moving this to the topic sentence of the paragraph

Methods:

Very detailed and well thought-out section. Study design is clearly explained.

Line 121-122: “and perceptions toward HTN in the context of HIV and integrated HIV-HTN care.” this reads awkwardly, consider rephrasing

Line 128: “three hospitals”: what type of hospitals - national, regional, district?

Line 140: rather than ‘we enrolled all PLHIV’, consider ‘we identified PLHIV’ through consecutive sampling. Because likely not all of those identified were enrolled, as likely some did not provide informed consent or agree to participate. Please present data of all the clients you identified as eligible, how many decided to enroll?

Line 161: how were the targeted sample sizes for each facility chosen?

Line 213: Figure 1 is terrific

Line 221: Table 3: Distance from home to clinic: Please specify the unit of measurement

Results: Comparing survey outcomes from clients at public vs. private facilities would be interesting as the integration models at these two facilities likely differ; consider adding this analysis.

Line 236-239: consider clarifying these two sentences as they seem to be a bit redundant.

Lines 249: “We identified patient needs and resources as a key barrier to integration of hypertension into HIV care.” I don’t understand the thinking here. Is the ‘patient need’ the need to have blood pressure checked very frequently? This is not consistent with evidence-based guidelines. What is the “patient resource” identified here? How do these needs (or resources?) act as a barrier to integration?

Line 254-257: How often should clients have their blood pressure checked to quality as ‘optimal hypertension care’? It’s unclear to me why this paper is suggesting that having a blood pressure checked within ‘the last seven days’ constitutes optimal care. I doubt the Ugandan national guidelines recommend blood pressure checks this frequently for hypetensive clients. Do these clients have home blood pressure cuffs?

Line 264: it would help to have some background on what the integration models are in these various care settings. For example, some ‘integrated’ models have hypertension screening in the ART clinic but then refer to NCD clinics or OPD for hypertension treatment. This client’s report of outside referrals for HTN care could be a critique of the integration model itself, rather than a critique that the HIV clinic is providing inadequate services.

Line 271: ‘exorbitant’ is subjective: would provide actual prices and show (rather than tell) that these prices are very high, or would state ‘private clinics allegedly sold these medications at prices that caused financial difficulties for patients”.

Line 315: “Wakiso district had a significantly higher score of 28.6 compared to Kampala district with a score of 26.1.” Was this difference statistically significant? If so, please provide a p value. If not, remove modifier ‘significantly’.

Line 317: same comment for ‘significantly’

Line 318: FIgure 3: I cannot tell which box plot corresponds to which score. Please label this figure more clearly and label Y axes. Only one of these 4 box plots has a significant P value - which score does this correspond to?

Discussion:

Line 374: “These findings suggest that strengthening and integrating HTN care services is critical to overcoming these barrier”. It seems to me that integration is but one answer for the patient level concerns you have identified. For example, ‘integration’ per se will not ensure that blood pressure medications are free or well-stocked at the HIV pharmacy. ‘Integration’ does not ensure that HIV clinical staff is well-trained in hypertension counseling, screening, and treatment. You need alternate funding models (increased government or donor support? National health insurance fund? Revolving fund pharmacies? etc), supply chain support for antihypertensives, increased support for HRH through training and task-shifting, hypertension screening and education in the community (ie not only facility-based) etc. Is hypertension treatment streamlined with simplified treatment algorhythms of a few antihypertensive medications and policies that permit task-shifting and nurse-prescribing? Overcoming these barriers is more complex than ‘strengthening integration’. Please be more nuanced about your recommendations. It is not accurate to suggest that an unspecified model of ‘integration’ will fix these barriers.

Line 431-432: reference?

Conclusion:

Line 437-438: “To address these gaps, health education for co-morbidities with HIV should be routinely integrated into HIV care.” This is a bit sparse for a conclusion. Perhaps circle back to how this study fits into the larger PULESA framework and what future and ongoing work consists of.

Reviewer #2: This paper has the potential to make an important contribution in the area of overlap in the need for care among adults/older adults living with HIV and hypertension (or other NCDs) in sub-Saharan Africa. The front end of the paper is very well written and clearly outlines the issues that need attention. The paper makes use of a unique and rich data source from Uganda that includes both survey data and qualitative interviews with individuals seeking care for HIV, who also have HTN (or are at risk for HTN). As is clearly pointed out, despite there being a policy for HIV-NCD care integration, the reality on the ground is that this is not the experience that most patients in these hospitals/clinics experience. Using a well-established framework Consolidated Framework for Implementation Research (CFIR), the authors aim to offer the experience of patients in these settings, and use these findings to outline recommendations.

The authors allude to benefits of the mixed-methods design that they use. Similar designs have been called - data-linked nested studies - and some of the ways to take advantage of the unique added value of the data are outlined here: Schatz E. 2012. Rationale and procedures for nesting semi-structured interviews in surveys or censuses. Population Studies 66(2): 183–95. https://doi.org/10.1080/00324728.2012.658851.

The findings and recommendations align well with other work from sub-Saharan Africa, and Uganda, that have looked at similar issues. However, the presentation of the findings - both quantitative and qualitative could be improved to make a stronger case for the underlying story or narrative that the authors are aiming to share. Currently the results part of the paper reads more like a report than an academic paper, with the data under analyzed and simply presented for the reader to interpret. The qualitative data are basically presented as anecdotal quotes rather than in a way that is convincing evidence that this is the patterns that emerged – more description of difference (across sites, by age, by gender, diagnosis, etc) or highlighting outliers could strengthen the qualitative results. The tables are hard to read and interpret. For example, Figure 3 is not described in the text in a way that explains what these box plots mean. What is a 'good' KAP score? What is meaningfully missing when a KAP score is not good?

There is additional literature on aging, HIV, NCDs in Uganda/SSA/LMICs that are not included in the references that might be useful - while it isn't necessary to include all of them, it did seem like some key authors on Uganda were missing - Seeley, Wandera, etc. Some of these are:

Coovadia, H., Jewkes, R., Barron, P., Sanders, D. and McIntyre, D. 2009. The health and health system of South Africa: historical roots of current public health challenges. The Lancet.

Droti, B. 2014. Availability of health care for older persons in primary care facilities in Uganda. PhD thesis, London School of Hygiene & Tropical Medicine, University of London, London.

Kuteesa, M. O., Seeley, J., Cumming, R. G. and Negin, J. 2012. Older people living with HIV in Uganda: understanding their experience and needs. African Journal of AIDS Research.

Negin, J., Nyirenda, M., Seeley, J. and Mutevedzi, P. 2013. Inequality in health status among older adults in Africa: the surprising impact of anti-retroviral treatment. Journal of Cross-cultural Gerontology.

Nnko, S., Bukenya, D., Kavishe, B. B., Biraro, S., Peck, R., Kapiga, S., Grosskurth, H. and Seeley, J. 2015. Chronic diseases in North-West Tanzania and Southern Uganda. Public perceptions of terminologies, aetiologies, symptoms and preferred management. PLOS ONE.

Rabkin, M., Kruk, M. E. and El-Sadr, W. M. 2012. HIV, aging and continuity care: strengthening health systems to support services for noncommunicable diseases in low-income countries. AIDS.

Schatz E, Seeley J, Negin J & Mugisha J. 2018. They “don’t cure old age”: Delays to health care access among older adults in rural Uganda. Ageing & Society.

Wandera, S. O., Kwagala, B. and Ntozi, J. 2015. Determinants of access to healthcare by older persons in Uganda: a cross-sectional study. International Journal for Equity in Health.

Minor issues:

In Table 3 - says distance to clinic - assume this is kms, but doesn't specify

6. PLOS authors have the option to publish the peer review history of their article (what does this mean?). If published, this will include your full peer review and any attached files.

**Do you want your identity to be public for this peer review?** For information about this choice, including consent withdrawal, please see our Privacy Policy.

Reviewer #1: No

Reviewer #2: No

---

## [Decision Letter · Decision Letter 1]

PGPH-D-24-01877R1

Harnessing HIV clinics to deliver integrated hypertension care for People living with HIV in Uganda: A formative mixed methods study.

Dear Dr. Ayebare,

Thank you for submitting your manuscript to PLOS Global Public Health. After careful consideration, we feel that it has merit but does not fully meet PLOS Global Public Health’s publication criteria as it currently stands. Therefore, we invite you to submit a revised version of the manuscript that addresses the points raised during the review process.

There are a few minor and one major revision to your manuscript - situating your findings within the literature in the field, that I hope you can consider in your revised copy. We look forward to your revised copy for journal's consideration

We look forward to receiving your revised manuscript.

Kind regards,

Jepchirchir Kiplagat, MPH

Academic Editor

Journal Requirements:

Additional Editor Comments (if provided):

While attempts have been made to revise this manuscript, the authors still need to refine the discussion section to situate the findings within the available literature in the field

Reviewers' comments:

Reviewer's Responses to Questions

**Comments to the Author**

1. If the authors have adequately addressed your comments raised in a previous round of review and you feel that this manuscript is now acceptable for publication, you may indicate that here to bypass the “Comments to the Author” section, enter your conflict of interest statement in the “Confidential to Editor” section, and submit your "Accept" recommendation.

Reviewer #1: (No Response)

Reviewer #2: (No Response)

2. Does this manuscript meet PLOS Global Public Health’s publication criteria? Is the manuscript technically sound, and do the data support the conclusions? The manuscript must describe methodologically and ethically rigorous research with conclusions that are appropriately drawn based on the data presented.

Reviewer #1: Yes

Reviewer #2: Yes

3. Has the statistical analysis been performed appropriately and rigorously?

Reviewer #1: Yes

Reviewer #2: Yes

4. Have the authors made all data underlying the findings in their manuscript fully available (please refer to the Data Availability Statement at the start of the manuscript PDF file)?

Reviewer #1: Yes

Reviewer #2: Yes

5. Is the manuscript presented in an intelligible fashion and written in standard English?

Reviewer #1: Yes

Reviewer #2: Yes

6. Review Comments to the Author

Reviewer #1: Re-reviewing Uganda HIV/HTN manuscript for PLOS Global Public Health

Thank you for your thorough response to my comments. Excellent work clarifying the integration models, the settings for clinical integration, and the differences between blood pressure outcomes by geography.

General: consider changing every instance of “patient” to “participant” to confirm with patient-centered language (https://www.niaid.nih.gov/research/hiv-language-guide).

Introduction:

Line 82: “However, in order to improve clinical outcomes (such as screening and blood pressure control), existing barriers to HTN and HIV integration must be addressed.” Screening is not a clinical outcome; would remove.

Line 110: “Prescription and dispensing of antihypertensive medications at the same pharmacy as antiretroviral therapy.” Not a complete sentence; please revise.

Line 112-119: I’m still a bit unclear on how your current study relates to PULESA- it is one of six studies within the larger PULESA NHLBI grant?

Line 356/Table 6: “Only 44% of patients reported having their had their blood pressure checked within the 357 last seven days with 24% being checked between eight and thirty days ago.” You mentioned in your response that ‘guidelines recommend two measurements per day for a minimum of three and ideally 7 days, with an average of the measurements being used to diagnose hypertension’ and that you used ‘BP checks within seven days as one proxy indicator for optimal care’. I do not see these recommendations in the 2020 Uganda HIV guidelines, nor do I see anything there about home blood pressure monitoring. Can you please provide the guidance that recommends this? Can you please clarify in the text, if this is indeed correct, that your participants had home blood pressure cuffs and also please provide the frequency with which they were advised to check their home BPs? I am not convinced that “BP checks within seven days” should be considered a proxy indicator for optimal care in this setting - nor in any setting. Many HTN SOPs in subSaharan Africa call for repeat BPs on a monthly basis until HTN control is achieved.

Line 496: The sections on Relative Advantage and Costs seem to be making the same argument- consolidate?

Line 497: “Participants identified several advantages to the integration of HTN and HIV services, particularly in overcoming the barriers of fragmented care and high costs associated with antihypertensive medications.” I understand how service integration can decrease the cost of travel to reach the clinic when clients are making fewer visits, but how does integrated service delivery reduce the high costs of antiHTN medications? Nothing in the text supports this.

Discussion:

Line 648 - 674: I would make this section its own paragraph, and try to really tighten up the writing. There are a few themes here that are mentioned repeatedly and should be condensed (decreased transportation costs, inconsistent medication supply, increased convenience, improved adherence). I would remove the 3 mentions of ‘stigma’ here because it’s beyond this scope- I don’t think your research touched on integrated models reducing stigma.

Reviewer #2: I still find the qualitative aspects of this work underanalyzed. The authors made efforts to change this section - not just listing quotes - however, it still has too many sub-sections, not enough information about any variation across age/sex/geography/health status, and not enough information about how common particular answers were and what types of answers clustered together.

I also find the discussion to continue to be insufficient in engaging with extant literature on aging in Uganda, HIV & chronic disease, and aging in underresourced health care environments.

The paper has the potential to provide important information on the lives and experiences of older Ugandans living with HIV, so I hope that the authors continue to revise to make the paper as strong as it has the potential to be.

7. PLOS authors have the option to publish the peer review history of their article (what does this mean?). If published, this will include your full peer review and any attached files.

**Do you want your identity to be public for this peer review?** For information about this choice, including consent withdrawal, please see our Privacy Policy.

Reviewer #1: **Yes: **Deborah Goldstein

Reviewer #2: No

---

## [Decision Letter · Decision Letter 2]

Harnessing HIV clinics to deliver integrated hypertension care for People living with HIV in Uganda: A formative mixed methods study.

PGPH-D-24-01877R2

Dear Ms Ayebare,

We are pleased to inform you that your manuscript 'Harnessing HIV clinics to deliver integrated hypertension care for People living with HIV in Uganda: A formative mixed methods study.' has been provisionally accepted for publication in PLOS Global Public Health.

Best regards,

Jepchirchir Kiplagat, Ph.D

Academic Editor

Reviewer Comments (if any, and for reference):

Reviewer's Responses to Questions

**Comments to the Author**

1. If the authors have adequately addressed your comments raised in a previous round of review and you feel that this manuscript is now acceptable for publication, you may indicate that here to bypass the “Comments to the Author” section, enter your conflict of interest statement in the “Confidential to Editor” section, and submit your "Accept" recommendation.

Reviewer #1: All comments have been addressed

Reviewer #2: All comments have been addressed

2. Does this manuscript meet PLOS Global Public Health’s publication criteria? Is the manuscript technically sound, and do the data support the conclusions? The manuscript must describe methodologically and ethically rigorous research with conclusions that are appropriately drawn based on the data presented.

Reviewer #1: Yes

Reviewer #2: Yes

3. Has the statistical analysis been performed appropriately and rigorously?

Reviewer #1: Yes

Reviewer #2: Yes

4. Have the authors made all data underlying the findings in their manuscript fully available (please refer to the Data Availability Statement at the start of the manuscript PDF file)?

Reviewer #1: Yes

Reviewer #2: Yes

5. Is the manuscript presented in an intelligible fashion and written in standard English?

Reviewer #1: Yes

Reviewer #2: Yes

6. Review Comments to the Author

Reviewer #1: Thank you for carefully and thoughtfully addressing these concerns; the paper is much improved and ready for publication.

Reviewer #2: This paper is much improved and I appreciate the care the authors took in restructuring the results and integrating literature on aging in Uganda.

7. PLOS authors have the option to publish the peer review history of their article (what does this mean?). If published, this will include your full peer review and any attached files.

**Do you want your identity to be public for this peer review?** For information about this choice, including consent withdrawal, please see our Privacy Policy.

Reviewer #1: No

Reviewer #2: No
